

# Response surface optimization and flavor determination of fermentation processes of orange peel tea wine

Yanbo Liu[1,2,3,4], Liu Mengge[1,2,3,4], Pengpeng Zhang[4,5], Wenxi Liu[1,2,3,4], Chong Yang[1,2,3,4], Jiayi Cui[1,2,3,4], Haideng Li[4,5] and Chunmei Pan[1,2,3,4]

[1] Henan Liquor Style Engineering Technology Research Center, Henan University of Animal Husbandry and Economy, Zhengzhou, China
[2] Henan Province Brewing Special Grain Development and Application Engineering Research Center, Henan University of Animal Husbandry and Economy, Zhengzhou, China
[3] Zhengzhou Key Laboratory of Liquor Brewing Microbial Technology, Henan University of Animal Husbandry and Economy, Zhengzhou, China
[4] College of Food and Biological Engineering (Liquor College), Henan University of Animal Husbandry and Economy, Zhengzhou, China
[5] College of Biological Engineering, Henan University of Technology, Zhengzhou, China

Corresponding author
Chunmei Pan, sige518888@163.com

## ABSTRACT

**Background:** In this study, Xinyang Maojian tea and orange peel were used as raw materials to brew health wine with orange and tea flavors.

**Methods:** Based on a single factor, the response surface method was used to optimize the fermentation of orange peel tea wine. Material volume ratio, yeast addition, fermentation temperature, fermentation time, and sucrose addition were used as the single-factor variables. The fermentation conditions of orange peel tea wine obtained by this method provide a theoretical basis for the development and utilization of tea wine and orange peel, which can promote the development of the tea wine market.

**Results:** The results showed that the material volume ratio of orange peel juice to tea juice was 1:3, the yeast addition amount was 4.9%, the fermentation temperature was 29 °C, the fermentation time was 7 d, and the sucrose addition amount was 29%. The main change indexes in the fermentation process of orange peel tea wine were determined, and the results were consistent with the fermentation law of orange peel tea wine. The contents of total flavonoids and total phenols in orange peel tea wine were 0.48 and 2.32 mg/mL, respectively, and were obtained using spectrophotometry and the Folin–Ciocalteu (F–C) method. The scavenging rate of 2,2-diphenyl-1-picrylhydrazyl (DPPH) radicals was 90.8%, and the scavenging rate of ·OH radicals was 77.3%. A total of 26 flavor compounds were detected by gas chromatography-mass spectrometry (GC-MS). The main aroma compounds were ethanol, 3-methyl-1-butanol, 2-methyl-1-propanol, phenylethyl alcohol, acetic acid, n-hexadecanoic acid, 2,4-di-tert-butylphenol, and other compounds.

**Conclusion:** The resulting orange peel tea wine was transparent, yellow in color, harmonious in flavor, and had certain health benefits, including strong antioxidant properties. The results of this study provide the theoretical basis for the research and development of tea wine.

## INTRODUCTION

China is the largest citrus grower in the world. Citrus fruits are mostly canned and juiced because of their color, flavor, and high nutritional value, but citrus fruit wine is a newer and rarer citrus product in China, and brewing with the citrus by-product orange peel is even more rare (*Li, Zhang & Qin, 2016*). Orange peel is rich in nutrients. Studies have shown that orange peel is rich in polymethoxy-flavone, which has anti-cancer, antioxidant, anti-mutagenic, and cardiovascular pharmacological effects. Polymethoxy-flavone can also control blood sugar, burn calories, and effectively control weight (*Xu et al., 2019*; *Neba Ambe et al., 2022*). The orange peel and other by-products comprise 40–50% of citrus fruits. These by-products are usually treated as waste, which has negative environmental effects and wastes resources (*Wang & Dan, 1999*). Orange peel, therefore, has broad application value.

Tea is a natural, traditional health drink. The polyphenols contained in tea have very strong antioxidant effects and physiological activity. These polyphenols are also a type of free radical scavenger, which can prevent the synthesis of various carcinogenic substances, such as nitrite amine, in the human body. China is a major producer of tea (*Hou et al., 2006*; *Kanwar et al., 2012*). Henan Province is famous for Xinyang Maojian, a type of green tea. Making tea wine from Xinyang Maojian can save production costs and open up a new market for tea sales.

Tea wine can be roughly divided into three types according to production process: sparkling tea wine, preparation tea wine, and fermented tea wine (*Liu, Huang & Liu, 2011*). Fermented tea wine is a type of low-alcohol tea wine prepared by water extraction or alcohol extraction with sucrose and yeast fermentation. *Huang et al. (2020)* mixed and fermented kiwi tea wine, using kiwi fruit as auxiliary material, which was rich in taste, tea polyphenols, and vitamin C. *Zhang (2020)* mixed and fermented tea wine, using sophora flowers as auxiliary material, to obtain tea wine with sophora flower aroma and health benefits. *Jin et al. (2021)* used navel orange juice as auxiliary material to mix and ferment selene-rich tea wine, which had a strong fruit flavor, unique taste, and refreshing body. Tea wine is gaining increased attention in beverage research. Tea wine has crystal clear appearance and a unique tea fragrance (*Shi et al., 2022*). The health benefits of tea wine are also increasing the drink's popularity among the increasingly health-conscious population (*Liang et al., 2020*).

Although different types of fermented tea wines have been reported, the brewing of orange peel tea wines, using orange peel and tea leaves as raw materials, has not been reported. In this study, Xinyang Maojian and orange peel were used as raw materials to brew health fruit wine with orange and tea flavors. The brewing process was optimized to obtain the best process formula, determine related indexes, and analyze the main aroma substances by gas chromatography-mass spectrometry (GC-MS), providing the details for a novel tea wine. Making compound fruit wine with orange peel and tea as raw materials can help decrease the waste of citrus industry by-products and promote the economic effect of tea in Henan province.

## MATERIALS AND METHODS

### Materials

Oranges, Xinyang Maojian, and white sugar were purchased from Zhengzhou Longhai market for this experiment. Cellulase was purchased from Shanghai Macklin Biochemical Technology Co., Ltd. (Shanghai, China); pectinase was purchased from Henan Runbu Biotechnology Co., Ltd.; β-cyclodextrin was purchased from Shanghai Macklin Biochemical Technology Co., Ltd.; angel wine yeast was purchased from Angel Yeast Co., Ltd. (Hubei, China); and chitosan was purchased from Beijing Boaotuo Technology Co., Ltd. Eight layers of gauze was used for filtration.

### Instruments

The experiment was conducted using a GCMS-QP2010 Ultra gas chromatography mass spectrometer (Shimadzu, Kyoto, Japan). An SW-CJ-2FD two-person, single-side purification table was purchased from Suzhou Purification Equipment Co., Ltd.; an HH-6 digital display constant temperature water bath was purchased from Changzhou Fangke Instrument Co., Ltd.; JXFSTPRP-24 grinding instrument was purchased from Shanghai Jingxin Technology Co., Ltd. (Shanghai, China); drying oven DHG-9075A was purchased from Shanghai Heng Scientific Instrument Co.; JW-2018HR medical centrifuge Anwei Jiawen Instrument Equipment Co.; 754 Series UV-Vis spectrophotometer Shanghai Yoke Instrumentation Co. (Shanghai, China).

### Methods

The operation points are as follows (as shown in Fig. 1):

(1) Miscellaneous removal and crushing: impurities and stems in the tea were removed and the remaining tea leaves were crushed to 3–5 mm;

(2) Tea extraction: 3.5% tea leaves were soaked in cold water for 30 min, then the tea leaves were removed from the water. The hot water bath method was used to extract the tea. Purified water with a pH of 4.5 (pH adjusted by 30% citric acid) (*Wu, She & Chai, 2000*) was extracted in a constant temperature water bath at 80 °C for 20 min, and the tea soup was cooled to be used.

(3) Orange peel preparation: orange peels without pests and mildew were chosen and removed from the fruit before cleaning. The orange peels were then baked at 95–100 °C for 2 min to remove excess essential oil and harmful miscellaneous bacteria.

(4) Crushing: after drying at 60 °C, orange peels were pulverized to 2–3 mm, and then 3.5% orange peel was extracted with water.

(5) Hydrolysis: 0.1% pectinase and 0.1% cellulase was added to the orange peel extract, which was then kept at a warm 45 °C temperature for 10 h and stirred evenly every hour.

(6) Filtration: after 20 min of micro-boiling treatment, the orange peel extract was filtered with double gauze, and the filtered residue was discarded.

(7) Removing bitterness: 0.3% β-cyclodextrin was added to remove bitterness, left for 12 h, and then the supernatant was removed (*Han et al., 2019*).

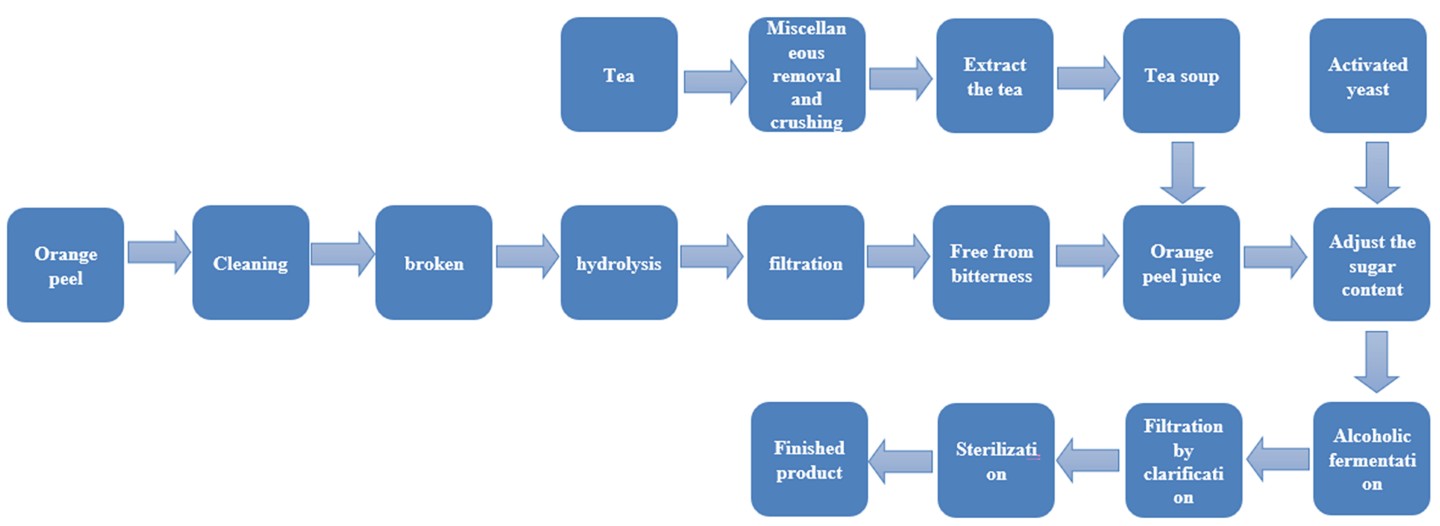

**Figure 1** Production process of orange peel tea wine.

(8) Adjusting the sugar content: the tea soup was mixed with the orange peel supernatant (3:1), and 25% white sugar was added.

(9) Activation of yeast: the yeast was added into 5% sugar water at a ratio of 1:20 (g/mL) and kept at 37 °C for 30 min to prepare the yeast activation solution.

(10) Alcohol fermentation: the yeast activation solution was added to the orange peel tea juice according to the inoculum amount of 0.8%. The tea juice was then capped and sealed and fermented at 28 °C for 7 d.

(11) Clarification and filtration: the fermented wine was filtered, and 0.2 g/L of 2% chitosan was added. After standing for 4 d, the supernatant was extracted for clarification and then filtered.

(12) Sterilization: After fermentation, the orange peel tea wine was filtered and kept in a 65 °C water bath for 30 min for pasteurization.

## Single optimization factor test of orange peel tea wine production process

The following were the baseline conditions of the wine production process: material volume ratio of orange peel juice to tea juice of 1:3, yeast addition amount of 0.8%, fermentation temperature of 28 °C, fermentation time of 7 d, and sucrose supplementation of 25%. For each single optimization factor test, only one factor was changed and the rest were kept unchanged. The following factors were individually investigated: material volume ratio of orange peel juice to tea juice (1:5, 1:4, 1:3, 1:2, 1:1, 2:1), yeast addition (0.8%, 1.6%, 2.4%, 3.2%, 4.0%, 4.8%, 5.6%, 6.4%), fermentation temperature (20 °C, 24 °C, 28 °C, 32 °C, 36 °C), fermentation time (3 d, 5 d, 7 d, 9 d, 11 d, 13 d, 15 d, 17 d, 19 d), and sucrose supplementation (15%, 20%, 25%, 30%, 35%, 40%). Sensory score and alcohol content were the main evaluation indexes. All tests were performed in triplicate.

## Response surface optimization experiment of orange peel tea wine production process

Based on the results of the single factor experiment, fermentation temperature (A), yeast addition (B), and sucrose addition (C) were selected as the three factors for the response surface optimization experiment. The response surface software Box-Behnken Design was used to study the best fermentation conditions of orange peel tea wine. Finally, a verification test was performed. The factors and levels of the response surface test are shown in Table 1.

## Sensory scoring method of orange peel tea wine

The sensory score was calculated according to the sensory scoring method outlined in Analytical Method of Wine and Fruit Wine (GB/T 15038-2006; *Brewing Sub-Technical Committee of National Food Industry Standardization Technical Committee, 2006*). According to this method, 0.2, 0.3, 0.4, and 0.1 weights were assigned to the appearance, smell, taste, and style of the orange peel tea wine, respectively (*Han et al., 2018*) (Table 2). Twelve professionals were asked to perform the sensory evaluations of the wine samples.

## Measuring main indicators of the orange peel tea wine fermentation process

Yeast was added to the orange peel tea leaf extract and fermented at 29 °C. In order to better regulate the fermentation process of orange peel tea wine, the total sugar, total acid, pH, and yeast count were determined at 1 d, 2 d, 3 d, 4 d, 5 d, 6 d, and 7 d of fermentation using the following methods:

Yeast count: orange peel tea wine was gradient diluted in sterile distilled water. The diluent was inoculated on YPD solid medium and incubated at 28 °C for 1 to 2 d, after which the yeast was counted. This analysis was performed in triplicate and the average value was taken.

pH: A precision pH meter was used to measure the pH.

Total sugars and acids: The general analytical method for wine and fruit (GB/T15038-2006) was used to measure total sugars and acids.

## Determination of total flavonoids and total phenols in orange peel tea wine

### Determination of total flavonoid content

Orange peel tea liquor was diluted to prepare the sample solution, and 0.2414 mg/mL rutin standard solution, 10% aluminum nitrate, 5% sodium nitrite, and 10% sodium hydroxide solution were prepared in reagent bottles for use.

The standard curve was determined according to the methods outlined by *Yang et al. (2020)*.

First, 1.0 mL 5% sodium nitrite solution was added to 1.0 mL of the precision suction sample solution, mixed, and then left for 6 min. Then, 1.0 mL of 10% aluminum nitrate solution was added, mixed, and left for 6 min before 10% sodium hydroxide solution was

**Table 1 Factors and levels of Box-Behnken test.**

| Level | (A) Temperature/°C | (B) Yeast added amount/(%) | (C) Sucrose addition/% |
|---|---|---|---|
| −1 | 24 | 4 | 25 |
| 0 | 28 | 4.8 | 30 |
| 1 | 32 | 5.6 | 35 |

**Table 2 Sensory scoring criteria for orange peel tea wine.**

| Item | Stand | Score (s) | Weighted value |
|---|---|---|---|
| Appearance | Clear and transparent, no precipitation | 85–100 | 0.2 |
| | Transparent with a small amount of precipitation | 75–84 | |
| | The color is dark and there is precipitation | <74 | |
| Smell | The wine is pleasant, with strong tea and orange aromas | 85–100 | 0.3 |
| | The wine is light, with a slight aroma of tea and orange | 75–84 | |
| | Light bodied wine without fruit aroma | 54–74 | |
| | No fragrance, bad smell | <54 | |
| Taste | The wine is full bodied with moderate orange flavor | 85–100 | 0.4 |
| | The main flavor is more harmonious, the body is more delicate, tea flavor | 75–84 | |
| | The main flavor is not coordinated, the body is thin, slightly bitter | <74 | |
| Style | Typical perfect, unique style | 85–100 | 0.1 |
| | Typical is more perfect, not elegant enough | 75–84 | |
| | Rough wine quality | <74 | |

added and mixed. Absolute ethanol, as the blank, was then added until the solution reached a total volume of 25 mL. The solution was then mixed and colored for 30 min. Absorbance was measured at a wavelength of 503 nm. Then, the concentration of total flavonoids in the samples was calculated according to the standard curve equation, and parallel experiments were performed.

### Determination of total phenol content

Folin-Ciocalteu (F-C) reagent: the dilution of the folin-phenol reagent was doubled.

Preparation of the standard curve: the standard curve was prepared according to the methods outlined by *Li et al. (2007)*.

Test sample processing: 1.0 mL orange peel tea wine was added to a 50.0 mL volumetric flask with 49 mL of water and shaken well.

Sample determination: 1.0 mL sample solution was added to 5.0 mL water, 1.0 mL F-C chromogenic agent, and 3.0 mL 7.5% sodium carbonate solution, respectively, to develop color. The absorbance of the samples was measured at 760 nm wavelength after 2 h of storage. Then, the concentration of total phenols in the samples was calculated according to the standard curve equation, and parallel experiments were performed.

## Determination of antioxidant capacity of orange peel tea wine

### DPPH free radical scavenging assay

According to the methods of *Wang et al. (2018)* the orange peel tea wine was centrifuged at 12,000 r/min and 4 °C for 5 min, and the supernatant was taken for testing; 0.05 mg/mL 2,2-diphenyl-1-picrylhydrazyl (DPPH) radical solution was configured. The sample was prepared and then left to react in the dark for 40 min. The absorbance was measured at 517 nm and calculated according to the following formula:

$$E(DPPH)(\%) = [1 - (A_I - A_{II})/A_{III}] \times 100$$

$A_I$: 2.0 mL of orange peel tea wine mixed with 2.0 mL of 0.05 mg/mL DPPH solution; $A_{II}$: 2.0 mL orange peel tea wine mixed with 2.0 mL absolute ethanol; $A_{III}$: 2.0 mL of 0.05 mg/mL DPPH mixed with 2.0 mL absolute ethanol.

### ·OH free radical scavenging assay

According to the methods of *Xia, Shuang & Yang (2021)* 6 mmol/LFeSO$_4$ solution, 6 mmol/L H$_2$O$_2$ solution, and 6 mmol/L salicylic acid solution were prepared. After preparing the sample, the reaction time was 10 min. Then, 2.0 mL of 6 mmol/L salicylic acid solution was added, the solution was shaken well, and its absorbance was measured at 510 nm after standing for 30 min.

$$E(\cdot OH)(\%) = [1 - (A_I - A_{II})/A_{III}] \times 100$$

$A_I$: 2.0 mL of 6 mmol/LFeSO$_4$ solution, 2.0 mL wine sample, 2.0 mL of 6 mmol/L H$_2$O$_2$ solution; $A_{II}$: 2.0 mL of 6 mmol/LFeSO$_4$ solution, 2.0 mL wine sample, 2.0 mL distilled water solution; $A_{III}$: 2.0 mL of 6 mmol/LFeSO$_4$ solution, 2.0 mL distilled water, 2.0 mL of 6 mmol/L H$_2$O$_2$ solution.

## Determination of main volatile components in orange peel tea liquor by GC-MS

### Sample preparation

A 5.0 mL sample of orange peel tea wine was moved to a 20 mL headspace bottle. Then, 2.0 g sodium chloride was added, and the mixture was shaken evenly. The wine sample was preheated at 60 °C for 5 min, the extraction head was inserted into the headspace port, and the adsorption was carried out at 60 °C for 30 min. The extraction head was then extracted and inserted into the GC injection port, and the thermal analysis was carried out at 220 °C for 5 min.

### GC-MS conditions

The chromatographic conditions were HP-FFAP (30 m × 0.25 mm × 0.25 µm). The shunt mode was "no shunt" with a flow rate of 1.21 mL/min. The injection port temperature was 240 °C. To heat the sample, the temperature was kept at 40 °C for 3 min, then the sample was heated at 5 °C/min to 80 °C without holding, followed by 8 °C/min to 230 °C for 7 min (*Feng et al., 2015*).

Mass spectrometry conditions were as follows: ionization mode was set to "(EI) ionization source," electron energy was 70 eV, interface temperature was 220 °C, and ion source temperature was 220 °C.

## Determination of physical and chemical indexes of orange peel tea liquor

The sensory score of the orange peel tea liquor was calculated according to the sensory scoring method outlined in the Analytical Method of Wine and Fruit Wine (GB/T 15038-2006).

## Data analysis and processing methods

Each experiment was repeated three times and the results were averaged. Origin Pro 12.1 software (OriginLab, Northampton, MA, USA) was used for data mapping, SPSS 20.0 software (SPSS, Chicago, IL, USA) was used for one-way ANOVA, and Design Expert 10.1 software (Stat-Ease, Minneapolis, MN, USA) was used for response surface design and result analysis.

# RESULTS

## Single factor experimental results of orange peel tea wine
### The effect of material volume ratio on the quality of orange peel tea wine

At 28 °C with 25% sucrose and 0.8% yeast, the volume ratio of orange peel juice to tea juice was tested at 1:5, 1:4, 1:3, 1:2, and 1:1. The sensory evaluation and the alcohol content evaluation of these ratios of orange peel juice to tea juice were performed after 7 d of fermentation.

As shown in Fig. 2, when the volume ratio of orange peel juice to tea juice was 1:3, the sensory score and alcohol content of orange peel tea wine were the highest. At this ratio, there was a good balance of tea aroma and orange aroma, the style of the tea wine was typical, and the tea wine was full-bodied. When the volume ratio of orange peel juice to tea juice was 1:5 and 1:4, the orange flavor was not obvious, the tea taste was rich and bitter, and the sensory score and alcohol level were both low. When the volume ratio of orange peel juice to tea juice was 1:2 and 1:1, the orange flavor was too rich, overpowering the tea flavor, the aroma was not balanced, and the sensory score was low. The results show that the quality of the orange peel tea wine was the highest with a volume ratio of orange peel juice to tea juice of 1:3.

### Influence of sucrose content on the quality of orange peel tea wine

At 28 °C, with a volume ratio of orange peel juice and tea juice set to 1:3, and 0.8% yeast addition, the amount of sucrose added was tested at 15%, 20%, 25%, 30%, 35%, and 40%. The sensory evaluation and the alcohol content evaluation of the orange peel tea wine with these sucrose addition amounts were performed after 7 d of fermentation.

As shown in Fig. 3, as the amount of sucrose added increased, both the sensory score and alcohol content increased, which may be because an appropriate amount of sucrose can reduce the astringency of tea. Sucrose is also conducive to the production of alcohol by

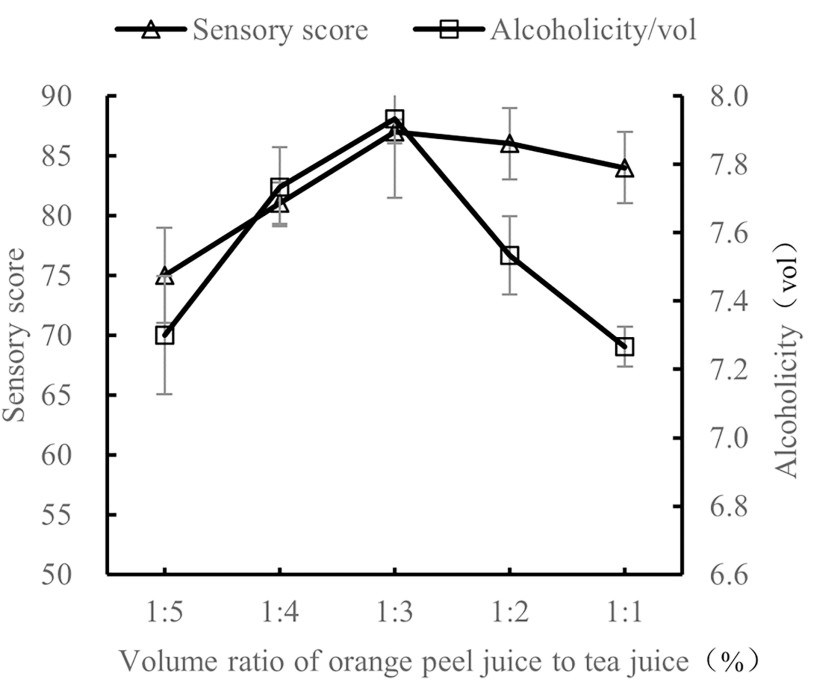

**Figure 2 Influence of material volume ratio on the quality of orange peel tea wine.**

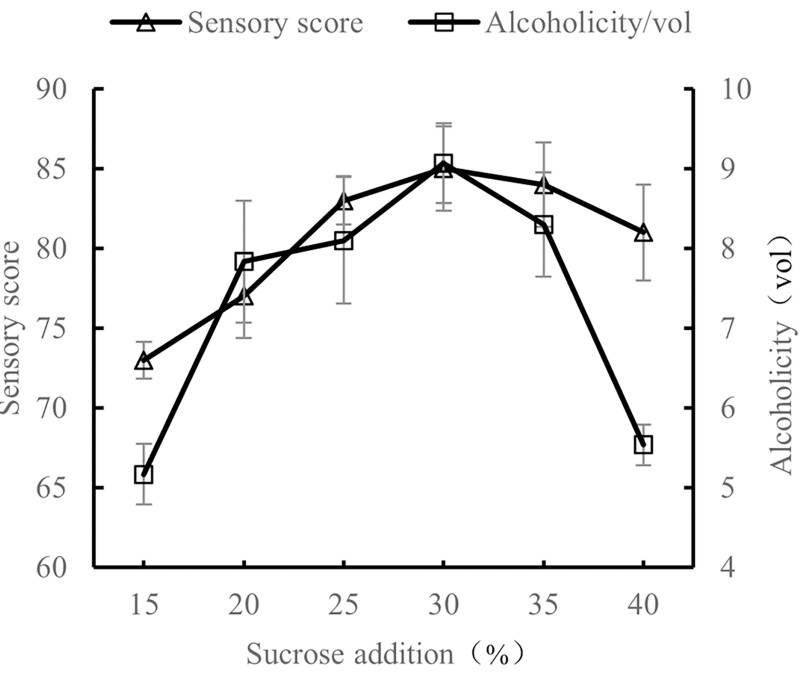

**Figure 3 Effect of sucrose content on the quality of orange peel tea wine.**

yeast. However, excessive sucrose made the orange peel tea wine too sweet and inhibited the activity of yeast, causing both the sensory score and alcohol content to decline. Excessive sucrose is also not conducive to the preservation of orange peel tea wine (*Han et al., 2018*; *Jia et al., 2023*). The results show that the quality of orange peel tea wine was the highest when sucrose content was 30%.

### Influence of fermentation temperature on the quality of orange peel tea wine

When the volume ratio of orange peel juice and tea juice was 1:3, the added amount of yeast was 0.8%, and the added amount of sucrose was 25%, the fermentation temperatures tested were 20 °C, 24 °C, 28 °C, 32 °C, and 36 °C. The sensory evaluation and the alcohol content evaluation of the orange peel tea wine were performed after 7 d of fermentation at the tested temperatures.

As shown in Fig. 4, alcohol content and sensory scores first increased and then decreased as fermentation temperature increased. Alcohol and sensory scores were highest at a fermentation temperature of 28 °C. When the temperature was too low (20 °C), yeast growth was slow, stunting the production of alcohol. When the temperature was too high (32 °C and 36 °C), lactic acid bacteria, acetic acid bacteria, and other microorganisms began to multiply, resulting in a sour orange peel tea wine taste (*Liu et al., 2022a*). The results show that the quality of orange peel tea wine was the highest when the fermentation temperature was 28 °C.

### Influence of fermentation time on the quality of orange peel tea wine

When the volume ratio of orange peel juice to tea juice was 1:3, the added amount of yeast was set to 0.8%, the added amount of sucrose was 25%, and the fermentation temperature was set to 28 °C, the fermentation times tested were 3 d, 5 d, 7 d, 9 d, 11 d, 13 d, 15 d, 17 d, and 19 d, after which the alcohol content and sensory evaluations were conducted.

As shown in Fig. 5, as fermentation time increased, alcohol content also increased before plateauing, and sensory scores first increased and then decreased. After 7 d of fermentation, sensory scores were the highest, with the wine having both a full taste and body. As fermentation time continued to increase, the wine took on a sour and astringent taste, and the sensory score decreased. The results show that the quality of fermented orange peel tea wine was the highest when the fermentation time was 7 d (*Liu et al., 2022b*).

### Influence of yeast addition amount on the quality of orange peel tea wine

When the volume ratio of orange peel juice and tea juice was 1:3, the added amount of sucrose was 25%, and the fermentation temperature was 28 °C, the amount of yeast added was tested at 0.8%, 1.6%, 2.4%, 3.2%, 4.0%, 4.8%, 5.6%, and 6.4%. The sensory evaluation and the alcohol content evaluation of the orange peel tea wine with these sucrose addition amounts were conducted after 7 d of fermentation.

As shown in Fig. 6, as the amount of added yeast increased, the alcohol and sensory scores also increased. When the yeast amount was 4.8%, sensory scores and alcohol scores were the highest. When the added amount of yeast was less than 4.8%, there was not enough yeast to make full use of sugar to produce alcohol, resulting in high residual sugar content and low alcohol content. When the added amount of yeast was more than 4.8%,

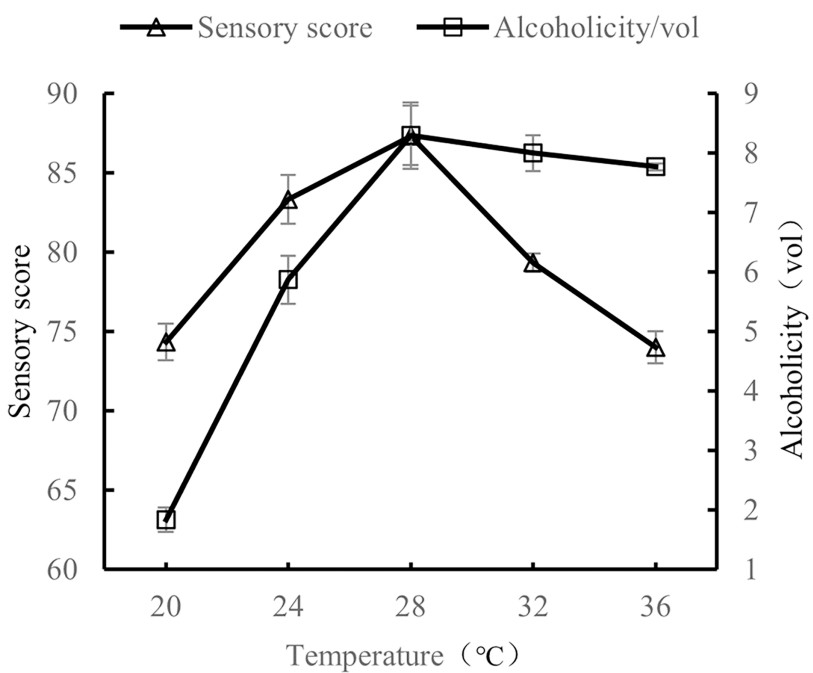

**Figure 4 Effect of fermentation temperature on the quality of orange peel tea wine.**

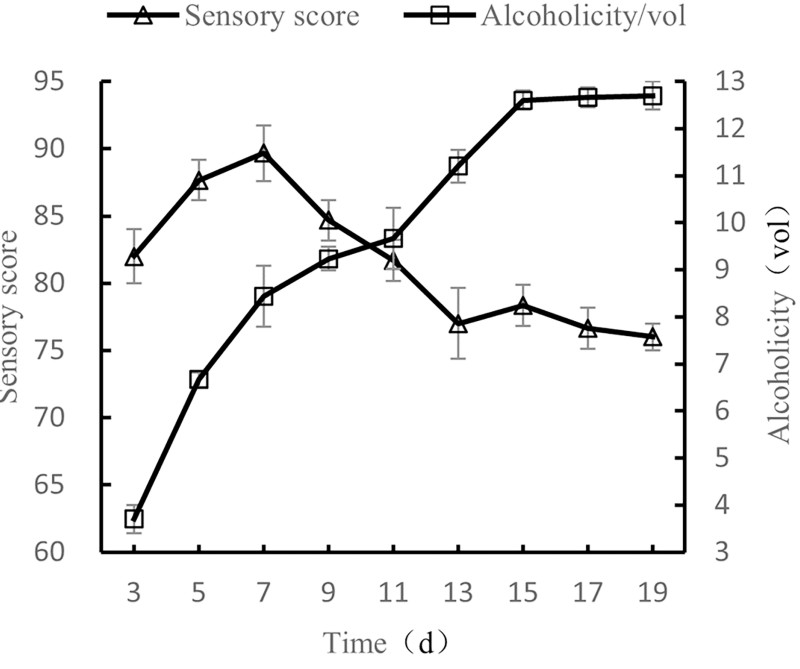

**Figure 5 Influence of fermentation time on the quality of orange peel tea wine.**

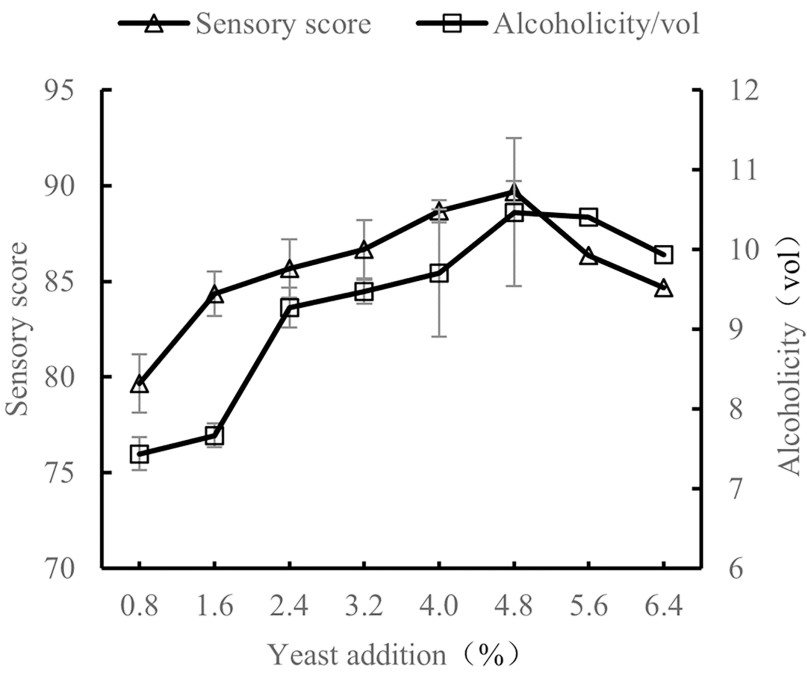

**Figure 6 Effect of yeast addition on the quality of orange peel tea wine.**

the added amount was too high, resulting in rapid consumption of nutrients and metabolic by-products, decreasing yeast metabolism and thus also decreasing alcohol content. Sensory scores also decreased with added yeast amounts higher than 4.8% (*Novo et al., 2014*). The results show the quality of orange peel tea wine was the highest when the yeast content was 4.8%.

## Response surface results and analysis of orange peel tea wine process optimization

### Response surface results of orange peel tea wine process optimization

After single factor testing, fermentation temperature (A), yeast content (B), and sucrose content (C) were selected to optimize the response surface of the brewing process of orange peel tea wine. The results of the response surface experiment are shown in Table 3, and the analysis of variance of the response surface experiment is shown in Table 4.

### Interaction analysis of influencing factors

As shown in Table 4, the F value of the quadratic regression model constructed by the response surface method was 60.64, and the effect was extremely significant ($P < 0.0001$). The lack of fit $P$-value was 0.2255, and the effect was not significant ($P = 0.2255 < 0.05$). These results indicate that all test sites could be described by this model. The determination coefficient $R^2 = 0.9873$ and the adjusted determination coefficient $R^2_{Adj} = 0.9711$ prove that the model had good reliability, and the regression model could predict the response value well. The primary term, A, and its secondary terms $A^2$, $B^2$, and

**Table 3 Box-Behnken design test scheme and results.**

| Experiment no. | (A) Temperature/°C | (B) Yeast added amount/(%) | (C) Sucrose addition/(%) | Alcohol content/%vol |
|---|---|---|---|---|
| 1 | 28.00 | 4.00 | 25.00 | 11.2 |
| 2 | 24.00 | 5.60 | 30.00 | 10.7 |
| 3 | 24.00 | 4.00 | 30.00 | 10.1 |
| 4 | 32.00 | 5.60 | 30.00 | 11.3 |
| 5 | 28.00 | 4.00 | 35.00 | 11.1 |
| 6 | 32.00 | 4.80 | 25.00 | 11.7 |
| 7 | 28.00 | 4.80 | 30.00 | 12.5 |
| 8 | 32.00 | 4.00 | 30.00 | 11.1 |
| 9 | 28.00 | 4.80 | 30.00 | 12.4 |
| 10 | 32.00 | 4.80 | 35.00 | 11.2 |
| 11 | 28.00 | 4.80 | 30.00 | 12.3 |
| 12 | 28.00 | 5.60 | 35.00 | 11 |
| 13 | 24.00 | 4.80 | 25.00 | 10.9 |
| 14 | 28.00 | 5.60 | 25.00 | 11.8 |
| 15 | 28.00 | 4.80 | 30.00 | 12.5 |
| 16 | 24.00 | 4.80 | 35.00 | 9.9 |
| 17 | 28.00 | 4.80 | 30.00 | 12.6 |

**Table 4 Analysis of variance of regression model.**

| Source | Sum of squares | Degrees of freedom | Mean square | F value | Prob > F value | Significance |
|---|---|---|---|---|---|---|
| Model | 10.88 | 9 | 1.21 | 60.64 | <0.0001 | ** |
| A | 1.71 | 1 | 1.71 | 85.87 | <0.0001 | ** |
| B | 0.21 | 1 | 0.21 | 10.60 | 0.0139 | * |
| C | 0.72 | 1 | 0.72 | 36.13 | 0.0005 | * |
| AB | 0.040 | 1 | 0.040 | 2.01 | 0.1995 | |
| AC | 0.063 | 1 | 0.063 | 3.14 | 0.1199 | |
| BC | 0.12 | 1 | 0.12 | 6.15 | 0.0423 | * |
| $A^2$ | 4.25 | 1 | 4.25 | 213.40 | <0.0001 | ** |
| $B^2$ | 1.81 | 1 | 1.81 | 90.64 | <0.0001 | ** |
| $C^2$ | 1.18 | 1 | 1.18 | 59.35 | 0.0001 | ** |
| Residual | 0.14 | 7 | 0.020 | | | |
| Lack of fit | 0.087 | 3 | 0.029 | 2.24 | 0.2255 | |
| Pure error | 0.052 | 4 | 0.013 | | | |
| Total | 11.02 | 16 | | | | |

**Note:**
One asterisk (*) or two asterisks (**) in the same row indicates a significant difference ($P < 0.05$).

$C^2$ are highly significant ($P < 0.01$); primary items B and C and interaction item BC were all significant ($P < 0.05$).

On the response surface, maximum values can be observed, indicating that the selected factor levels are reasonable. The contours of fermentation temperature (A) and yeast

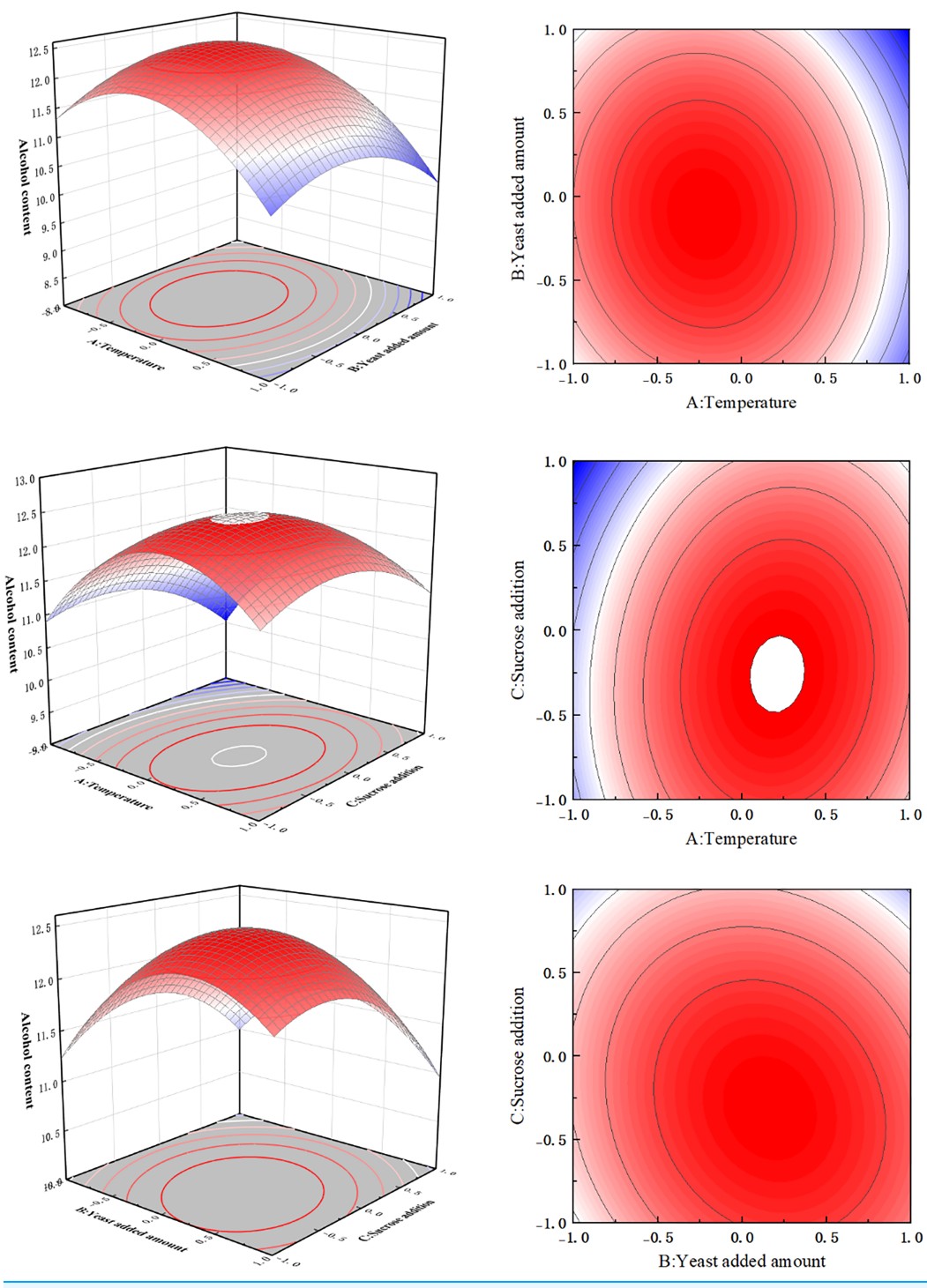

**Figure 7 Response surface diagram.**

addition (B), and fermentation temperature (A) and sucrose addition (C) were approximately circular, indicating that the effect on the alcoholic strength of orange peel tea wine was not significant (Fig. 7). Whereas, the contours between yeast addition (B) and

sucrose addition (C) were elliptical, indicating that these two factors had significant effects on the alcohol content of orange peel tea wine. According to the response surface software analysis, the optimal fermentation conditions for the brewing process of orange peel tea wine were as follows: fermentation temperature 28.82 °C, yeast content 4.92%, and sucrose content 28.58%. For simplicity in production, these conditions were appropriately modified to a fermentation temperature of 29 °C, a yeast content of 4.9%, and a sucrose content of 29%. Under these fermentation conditions, three flat tests were conducted, and the average alcohol content of orange peel tea wine was 12.56% vol, which was only 0.1% vol different from the theoretical predicted value, which verified that the response surface test data were accurate and reliable.

## Changes of main indexes in the fermentation process of orange peel tea wine

Changes in the yeast number, total sugar, pH, and total acid during the fermentation process of orange peel tea wine are shown in as Figs. 8, 9, 10, 11 respectively. The yeast number increased gradually as fermentation time increased, reaching a maximum on the 6th day before decreasing, which is consistent with the growth law of yeast. The change of total sugar content was inversely related to total yeast content. As yeast content increased, total sugar consumption gradually increased, lowering the total sugar content. During the period of rapid yeast growth, total sugar decreased rapidly, and in the later period, total sugar decreased slowly, which may be because the accumulation of alcohol inhibits yeast sugar consumption (*Zhang et al., 2015*; *Ekumah et al., 2021*). As fermentation time increased, the total acid first increased and then decreased, while the pH first decreased and then increased. When the total acid increased, the pH decreased. The acid produced by yeast in the fermentation process is the aromatic substance of wine, and increases the antibacterial effect of orange peel tea wine (*Jiang et al., 2014*).

## Determination of total flavonoids and total phenols in orange peel tea wine

Standard curves were drawn with the concentration of rutin standard solution and gallic acid as the horizontal coordinate and absorbance as the vertical coordinate, respectively. The measured regression equations were divided into $y = 3.5536x + 0.1951$ ($R^2 = 0.9900$), $y = 0.1016x + 0.0297$ ($R^2 = 0.9910$). The total flavonoid content and total phenol content were 0.48 and 2.32 mg/mL, respectively, when the absorbance of the samples was put into the standard curve (Table 5).

## Identification of antioxidant capacity of orange peel tea wine

DPPH free radical scavenging rate reached 90.8% and ·OH free radical scavenging capacity reached 77.3% (Table 6). These results indicate that orange peel tea wine has a good antioxidant capacity, which may be due to the contribution of total phenols and total flavonoids (*Ekumah et al., 2021*).

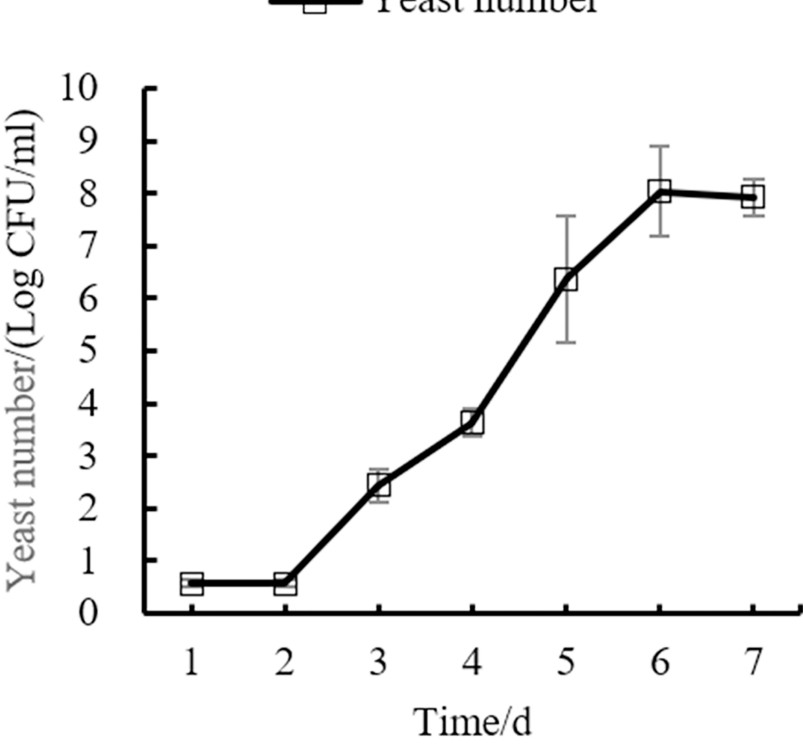

**Figure 8  Yeast number change.**

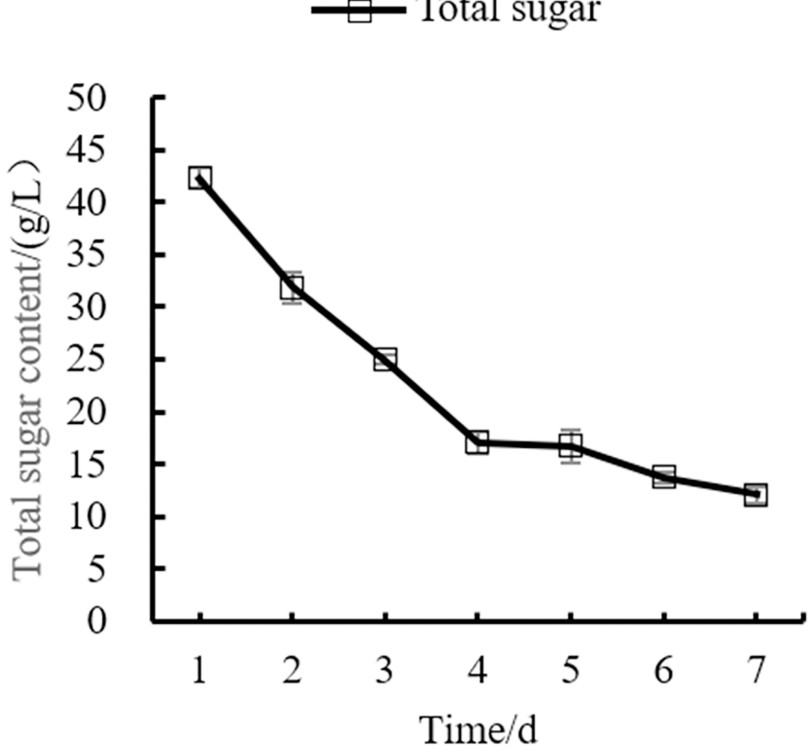

**Figure 9  Total sugar content change.**

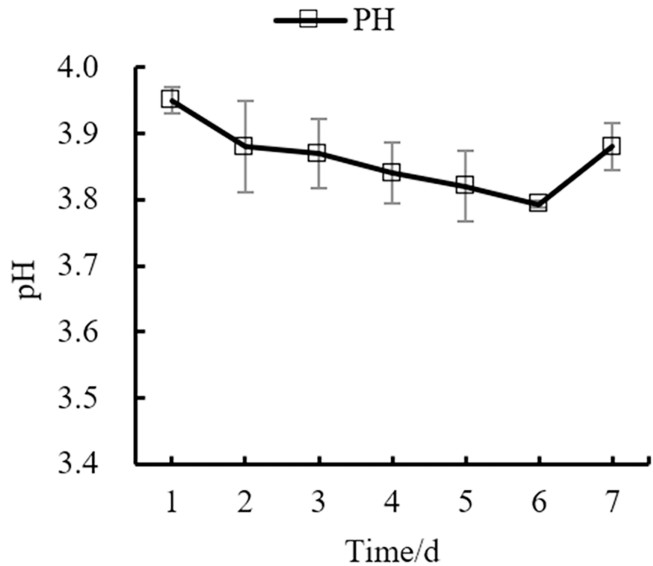

**Figure 10  pH change.**               

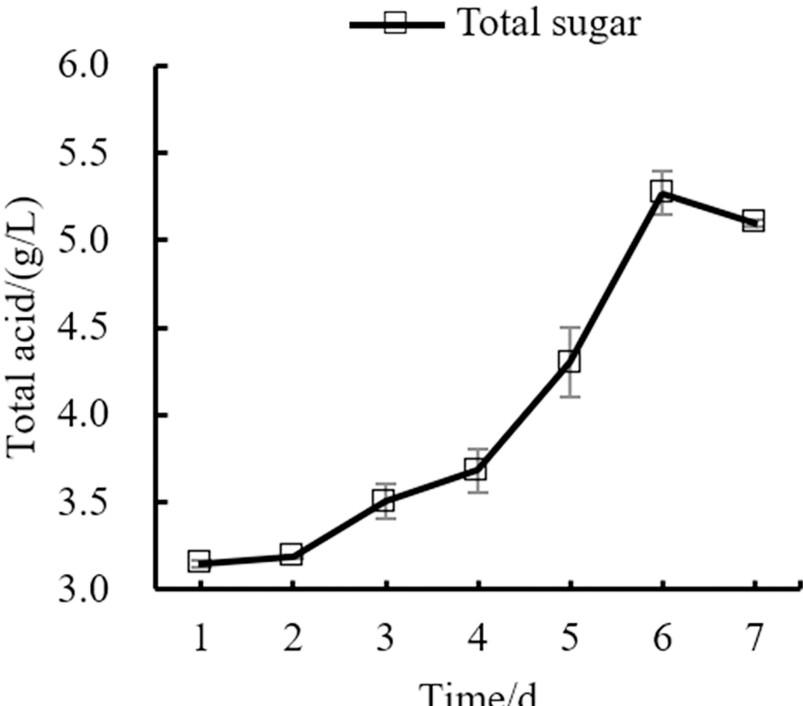

**Figure 11  Total acid content change.**     

## Analysis of aroma components of orange peel tea wine

Through GC-MS, a total of 26 flavor compounds of orange peel tea wine were identified, which mainly included alcohols, acids, esters, phenols, ketones, and other compounds (Table 7). Alcohols included 2-methyl-1-propanol, ethanol, 1-propanol,

**Table 5  Contents of total phenols and total flavonoids.**

| Name | Content |
|---|---|
| Total flavone (mg/mL) | 0.48 ± 0.05 |
| Total phenol (mg/mL) | 2.32 ± 0.06 |

**Table 6  Scavenging ability of DPPH free radical and ·OH free radical.**

| Name | Content |
|---|---|
| DPPH (%) | 90.8 ± 0.15 |
| ·OH (%) | 77.3 ± 0.20 |

**Table 7  Aroma components of orange peel tea wine.**

| | Number | Retention time/min | Compound | Relative content/% |
|---|---|---|---|---|
| Alcohols | 1 | 1.334 | 2-Methyl-1-propanol | 12.59 |
| | 2 | 3.010 | Ethanol | 44.77 |
| | 3 | 4.070 | 1-Propanol | 0.01 |
| | 4 | 6.846 | 3-Methyl-1-butanol | 31.53 |
| | 5 | 16.634 | (R)-4-Methyl-1-(1-methylethyl)-3-Cyclohexen-1-ol | 0.11 |
| | 6 | 17.932 | Alpha-terpineol | 0.20 |
| | 7 | 19.745 | (Z)- 3,7-Dimethyl-2,6-octadien-1-ol | 0.10 |
| | 8 | 20.544 | Phenylethyl alcohol | 2.11 |
| Acids | 9 | 14.269 | Acetic acid | 1.90 |
| | 10 | 16.158 | 2-Methyl-propanoic acid | 0.32 |
| | 11 | 17.563 | 2-Methyl-butanoic acid | 0.21 |
| | 12 | 19.425 | 2-Phenylethyl ester acetic acid | 0.06 |
| | 13 | 19.674 | Hexanoic acid | 0.16 |
| | 14 | 21.919 | Octanoic acid | 0.55 |
| | 15 | 23.933 | n-Decanoic acid | 0.48 |
| | 16 | 30.685 | n-Hexadecanoic acid | 1.02 |
| Esters | 17 | 25.467 | 6-Octadecenoic acid, (Z)-methyl ester | 0.05 |
| | 18 | 25.886 | 10,13-Octadecadienoic acid, methyl ester | 0.09 |
| Aldehydes | 19 | 19.475 | 2,4-Dimethyl-benzaldehyde | 0.28 |
| Phenols | 20 | 24.305 | 2,4-Di-tert-butylphenol | 1.43 |
| Ketones | 21 | 23.403 | 4-Hydroxy-2-methylacetophenone | 0.13 |
| Others | 22 | 2.314 | Hydrazinecarboxamide | 0.91 |
| | 23 | 19.300 | 3-O-methyl-d-fructose | 0.11 |
| | 24 | 25.149 | 2,3-Dihydro-Benzofuran | 0.09 |
| | 25 | 19.609 | Anethole | 0.11 |
| | 26 | 27.127 | Caffeine | 0.68 |

3-methyl-1-butanol, (R)-4-methyl-1-(1-methylethyl)-3-cyclohexen-1-ol, alpha-terpineol, (Z)-3,7-dimethyl-2,6-Octadien-1-ol, and phenylethyl alcohol. Acids included acetic acid, 2-methyl-propanoic acid, 2-methyl-butanoic acid, 2-phenylethyl ester acetic acid, hexanoic acid, octanoic acid, n-decanoic acid, and n-hexadecanoic acid. Esters included 6-Octadecenoic acid, (Z)-methyl ester,10,13-octadecadienoic acid, and methyl ester. These three types of substances accounted for more than 95% of the volatile aroma components.

The main aroma substances in orange peel tea wine were ethanol, 3-methyl-1-butanol, 2-methyl-1-propanol, phenylethyl alcohol, acetic acid, n-hexadecanoic acid, 2,4-di-tert-butylphenol, and other compounds. Phenylethyl alcohol and 2-methyl-1-propanol are the characteristic flavor components of green tea and orange peel, and their presence endows orange peel tea wine with the aroma of apple brandy and flowers, and alpha-terpineol is the main component of citrus aroma compounds, giving orange peel tea wine a citrus aroma (*Yin et al., 2021*; *Chen et al., 2019*). The presence of acids such as acetic acid, 2-methyl-propanoic acid, and octanoic acid, harmonizes the wine body and gives the tea wine an elegant aroma. Caffeine can give tea wine a special taste and is an important flavor component of tea wine (*Qiu et al., 2011*).

## CONCLUSION

Response surface optimization of orange peel tea wine fermentation was carried out using single-factor response surface methodology. The optimal fermentation conditions were as follows: material ratio of orange peel juice to tea juice of 1:3, yeast content 4.9%, fermentation temperature 29 °C, sucrose content 29%, fermentation time 7 days. Under these conditions, the orange peel tea wine had a unique taste, a typical style, the best sensory score, an alcohol content of 12.46% vol, 12.1 g/L of total sugar, 5.1 g/L of total acid, and a pH of 3.88. The main change indexes in the fermentation process of orange peel tea wine were determined, and the results were consistent with the fermentation law of orange peel tea wine. The total flavonoid content of orange peel tea wine was 0.48 mg/mL, the total phenol content was 2.32 mg/mL, the DPPH free radical clearance rate was 90.8%, the ·OH free radical clearance rate was 77.3%, and the antioxidant ability was good. Through gas chromatography-mass spectrometry, 26 flavor compounds were identified, with the main ones being ethanol, 3-methyl-1-butanol, 2-methyl-1-propanol, phenylethyl alcohol, acetic acid, n-hexadecanoic acid, 2,4-di-tert-butylphenol, and other compounds. The fermentation conditions of orange peel tea wine obtained in this study provide the theoretical basis for the development and utilization of tea wine and orange peel, and can promote the development of the tea wine market.

### Funding

This work supported by the Key Technologies Research and Development Program of Henan Province of China (202102110130), Major Science and Technology Projects of Henan Province of China (181100211400), Food Science and Engineering Key Discipline Construction Project of Henan University of Animal Husbandry and Economy (XJXK202203) and R&D and Demonstration Application of Key Technologies for

Ecological Brewing of Henan Baijiu (231111112000). The funders had no role in study design, data collection and analysis, decision to publish, or preparation of the manuscript.

## Grant Disclosures

The following grant information was disclosed by the authors:

Key Technologies Research and Development Program of Henan Province of China: 202102110130.

Major Science and Technology Projects of Henan Province of China: 181100211400.

Food Science and Engineering Key Discipline Construction Project of Henan University of Animal Husbandry and Economy: XJXK202203.

R&D and Demonstration Application of Key Technologies for Ecological Brewing of Henan Baijiu: 231111112000.

## Competing Interests

The authors declare that they have no competing interests.

## Author Contributions

- Yanbo Liu conceived and designed the experiments, performed the experiments, authored or reviewed drafts of the article, and approved the final draft.
- Liu Mengge analyzed the data, prepared figures and/or tables, and approved the final draft.
- Pengpeng Zhang analyzed the data, authored or reviewed drafts of the article, and approved the final draft.
- Wenxi Liu analyzed the data, prepared figures and/or tables, and approved the final draft.
- Chong Yang analyzed the data, prepared figures and/or tables, and approved the final draft.
- Jiayi Cui analyzed the data, prepared figures and/or tables, and approved the final draft.
- Haideng Li performed the experiments, authored or reviewed drafts of the article, and approved the final draft.
- Chunmei Pan conceived and designed the experiments, authored or reviewed drafts of the article, and approved the final draft.

## Data Availability

Raw data is available as a Supplemental File.

## Supplemental Information

Supplemental information for this article can be found online at http://dx.doi.org/10.7717/peerj.19357#supplemental-information.

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
