# Peer review of "Response surface optimization and flavor determination of fermentation processes of orange peel tea wine"

_PeerJ, doi:10.7717/peerj.19357_

## Round 0.1 · original submission · Major Revisions

As the authors can see, reviewers suggest a revision of the manuscript. I agree with their observations, and so I ask you to evaluate the comments and take the necessary actions.

·

Basic reporting

Topic is important and is focussed on resource recycling and value addition

Experimental design

Seems OK!

Validity of the findings

Seems fine, for further details, please see the comments below

Additional comments

The authors covered an important topic in developing a fermentation-based technique for resource conservation, recycling, and value addition. It can be considered for publication in PeerJ, however, manuscript is not well-written, authors should consider the following suggestions and should prepare a revised version.
1. Language needs improvement
2. Manuscript layout, and storytelling should be improved (write it like an interesting story)
3. Methods are written in an instruction manual style. This is not the proper format for a research paper, please refer to some previously published papers in PeerJ to learn more about the writing style of methods
4. Line 106: “Alcohol fermentation: yeast activation solution was added”. The fermenting microbes in the wine/liquor-making process are a complex consortium, which are selected over a longer time for a specific fermentation process. How did you manage this aspect? Besides, the authors did not mention whether it was a consortium. What kind of yeasts? What was the source of this inoculum?
5. Fermentation time ranged up to 19 days. How did you set that range? Why not 30 or 60 days? Commercial liquor making usually has different fermentation duration
6. Conclusion is repeating results. This section should contain what you conclude based on the results and findings of this study.
7. Add some challenges, issues of the study, and recommendations in the conclusion section
8. Fig. 1: the schematic diagram does not look nice, its presentation should be improved
9. Fig. 2 & 3 and others: units on the x and y coordinates should be written in brackets; For instance: Volume ratio of orange peel juice to tea juice (%), Temperature (°C), Time (days)
10. Please try to add some more recent references from 2023, 2024.

·

Basic reporting

The authors of this manuscript presented scientific results on an important issue of byproduct valorization (orange peel). They presented the manuscript as:
Unambiguous, professional English is used throughout the manuscript.
The literature is well-referenced, and sufficient field background/context is provided.
Professional article structure, figures, tables. Raw data shared.
Self-contained with relevant results to hypotheses.
However, I suggest that the title be reframed as:
Response surface optimization of fermentation process and determination of aroma components of wine from extracts of orange peel and tea blends

Experimental design

The experimental design is well organized.
Original primary research within Aims and Scope of the journal is presented.
The research question well defined, relevant & meaningful. It is stated how research fills an identified knowledge gap.
Rigorous investigation performed to a high technical & ethical standard.
Methods described with sufficient detail & information to replicate.
However, there are some informations missed such as:
Were the instruments (for example: GCMS-QP2010Ultra) bought for this research only?
Better to say GCMS-QP2010Ultra (origin from Shimadzu, Japan) was employed for the experimental work.
What filtration method/ equipment was applied?
The procedure for the determination of total phenols is incomplete.
Against which standard were the total phenols measured?
The procedure for the antioxidant capacity determination is incomplete.
Against which standards was the antioxidant capacity measured?
The fonts and alignments of the following title should be corrected.
2.8 Determination of main volatile components in orange peel tea liquor by GC-MS
2.8.1 Sample preparation
The method employed for the determination of the main volatile components in orange peel tea liquor by GC-MS is not mentioned. Optimized experimental conditions should be mentioned if the authors used this new method.

Validity of the findings

Meaningful replication is encouraged where rationale & benefit to literature is clearly stated.
All underlying data have been provided; they are robust, statistically sound, & controlled.
Conclusions are well stated, linked to the original research question & limited to supporting results.

However, validities of the developed models are strictly required.
The authors should work at least three trial experiments at the proposed optimum input variables (confirmation experiment) and validate the variance with the expected results using a statistical significance test such as a t-test. Moreover, evaluating the validity of the RSM models by P-values is not enough as the number of input variables increases, then the P-values will be affected. Hence, applying additional validation techniques like percent of absolute average deviation (%AAD) are important.

·

Basic reporting

The first section is not the Background, it's the Abstract. In the content of the same, there should be no sections, it should be written continuously. The parameters of the independent variables must be defined. The response surface design used should be indicated, as there is no design based on simple factors. Avoid ambiguous expressions such as: "... and has certain health function", should be written in quantitative form to which it refers.

The technical writing of the methodology section must be significantly improved; among the comments received are:

Line 86, indicate how the impurities were removed and of what type, indicate the equipment with which the grinding was carried out, including brand and model, as well as the size of the particle considered.
Line 89, specify the concentration of the citric acid used
Line 94, indicate the equipment used for drying and pulverizing, including model and brand
Line 100, specify the amount of β-cyclodextrin and the time it was allowed to rest
Line 108: Specify how and with what the filtration was carried out
Line 110: Specify how and with what the filtration was carried out, as well as the equipment used for the water bath
Rewrite from line 85 to 111, in one or two paragraphs, ensuring a smooth writing style to describe the methodology
Section 2.5
From the results obtained in sections 2.3 and 2.4, please specify the operational conditions used for this monitoring Additionally, indicate the quantity of orange peel tea leaves extract added
Rewrite this information in a coherent narrative, consolidating it into a single paragraph Include details on the amount of orange peel tea wine used and the dilution ratio Also, specify the incubation equipment, including its model and brand Lastly, provide the model and make of the potentiometer used, along with the reference for the “General Analytical Method for Wine and Fruit Wine”

Section 2.7.1
It must include the name, model and brand of the equipment with which the centrifugation was carried out.
Indicate the name, model and brand of the equipment where the absorbance was measured.
Lines 182 – 185 show "AI: 2.0 mL 6 mmol/LFeSO4 solution, 2.0 mL wine sample, 2.0 mL 6 mmol/L H2O2 solution; AII: 2.0 mL 6 mmol/LFeSO4 solution, 2.0 mL wine sample, 2.0 mL distilled water solution; AIII: 2.0 mL 6 mmol/LFeSO4 solution, 2.0 mL distilled water, 2.0 mL 6 mmol/L H2O2 solution", indicate if it is a mixture, how and where they are mixed at what temperature and the concentrations of the solutions, if it is always the same, then E(· OH)(%) is always a constant.
Phrases in lines 186 and 187 should be in bold
Section 2.8.2
The characteristics (brand, model, country of construction) of the chromatograph and mass spectrophotometer must be included
Section 2.9
The bibliographic reference for the general analytical method for wine and fruit wine must also be included

References
Approximately 47% of the included articles are older than 5 years Please update the references to reduce this to less than 30%

Figure 1 is not cited in the text

Overall, all figures should significantly improve their quality

Experimental design

Section 2.2 does not clearly explain the way they formulated the Single optimization factor This concept is associated with multi-objective problems, considering Pareto optimal solutions Therefore, it should present the constraint functions, the objective function, and include a table indicating the combination of conditions taken for each test For the tea juice, there are 6 levels; for yeast addition, 8 levels; for fermentation time, 9 levels; and for sucrose supplementation, 6 levels

In Section 2.3, each of the proposed levels for the BB design variables should be included Also, specify which variable is the response variable

Section 3.2.1
Table 4, include for each source include the % contribution and the value of p
Section 3.2.2
To complete the BB design, you will need to include the regression polynomial, canonical analysis to determine the stationary point, and the predicted response.
The discussion of the results obtained against other research should be carried out in order to assess the contribution achieved

Validity of the findings

RESULTS
Section 3.1.1
In section 2.2, six levels of study are indicated, the 2:1 level is missing
Figure 2 should indicate the range of deviation between the three samples analyzed at each point
Section 3.1.2
Figure 3 should indicate the range of deviation between the three samples analyzed at each point
Section 3.1.3
Figure 4 should indicate the range of deviation between the three samples analyzed at each point
Section 3.1.4
Figure 5 should indicate the range of deviation between the three samples analyzed at each point
Section 3.1.5
The levels indicated in line 162 do not coincide with the levels indicated in line 117
Figure 6 should indicate the range of deviation between the three samples analyzed at each point

Section 3.3
In line 306, Figures 8-11 are indicated, but only Figures 8 and 9 are included
In the description of the results in this section, numerical values should be included and compared against other research in order to assess the contribution achieved
Section 3.4
The discussion of the results obtained against other research should be carried out in order to assess the contribution achieved
Section .35
The discussion of the results obtained against other research should be carried out in order to assess the contribution achieved

MANDATORY All results obtained must be discussed against other research from JCR journals in the last five years

Conclusions
In line 349, it mentions “..was optimized by single factor..”, varying only one factor is not an optimization, an error that has been evident since section 2.2 It is recommended to consult Montgomery's book to properly indicate the analysis conducted

Additional comments

The research shows some relevance; however, the lack of technical structure in the presentation of the methodology and the absence of discussion of the results diminish its significance

The methodology presentation should be substantially improved to be more technically sound with a coherent and appropriate narrative

The quality of the figures must also be significantly enhanced

It is important to include a robust discussion of the experimental design, which is currently addressed only briefly

A more rigorous literature review is necessary to incorporate solid discussions that highlight the obtained results

---

## Round 0.2 · Major Revisions

Please check the reviewers' comments and prepare the necessary adjustments as well as a response to them.

·

Basic reporting

Seems OK now!

Experimental design

Well-defined

Validity of the findings

Justified

Additional comments

Authors have improved their manuscript.

·

Basic reporting

The authors reported clear existing facts. The developed illustrative figures, tables, and clear structure of the article.
Some comments are included in the manuscript as a track.
Correct the folin phenol method.
Do you mean Folin–Ciocalteu (F–C) method?
Activation of yeast
What specific yeast strain was used?
folinphenol reagent
Use the common name of the reagent solution
Fig. 8-11
Make these in 1 figure caption (as Fig. 8) and list them as 8 to 11

Experimental design

The authors developed and reported a clear experimental design and method of analysis.

Validity of the findings

Overall optimum values with their confirmation experimental results are very required from an optimization experiment.
Moreover, the statistical analysis of the predicted and measured values at these confirmation experimental is important. Then, all the final wine analyses made at the optimum values should be discussed.

Additional comments

The authors should work on a confirmation experiment and report the statistical analysis by comparing the expected and measured results.
Statistical models developed from the optimization process should be reported.

·

Basic reporting

I thank the authors for addressing some of the requests made in the first review; however, there are many that were not addressed and it is important that a response is provided. It is also requested that in the new extensive version, the modifications be marked in a different color and that a document be included where responses to the observations made are provided.

The abstract must be written in a single paragraph; there should be no sections (Background, Methods, Results, Conclusions). The response variable(s) that are established are not clearly indicated.

Experimental design

Line 102, indicate how the impurities were removed and of what type, indicate the equipment with which the grinding was carried out, including brand and model, as well as the size of the particle considered.

Line 112, indicate the equipment used for drying and pulverizing, including model and Brand
Line 131: Specify how and with what the filtration was carried out, as well as the equipment used for the water bath

Rewrite from line 101 to 132, in one or two paragraphs, ensuring a smooth writing style to describe the methodology
Section 2.3 You must include the levels used for each variable in the proposed Box-Behnken design.
Section 2.7.1
It must include the name, model and brand of the equipment with which the centrifugation was carried out.
Indicate the name, model and brand of the equipment where the absorbance was measured.
Lines 205 – 207 show " AI: 2.0 mL orange peel tea wine, 2.0 mL of 0.05 mg/mL DPPH solution; AII: 2.0 mL orange peel tea wine, 2.0 mL absolute ethanol; AIII: 2.0 mL of 0.05 mg/mL DPPH, 2.0 mL absolute ethanol ", indicate if it is a mixture, how and where they are mixed at what temperature and the concentrations of the solutions, if it is always the same, then E(· OH)(%) is always a constant. In line 205, “after 2.0 mL of…” indicate the sustance
Section 2.8.2
The characteristics (brand, model, country of construction) of the chromatograph and mass spectrophotometer must be included
Section 2.9
The bibliographic reference for the general analytical method for wine and fruit wine must also be included

Validity of the findings

RESULTS
Section 3.1.1
In section 2.2, six levels of study are indicated, the 2:1 level is missing
Figure 2 should indicate the range of deviation valor between the three samples analyzed at each point and incluide the 2:1 level
Section 3.1.2
Figure 3 should indicate the range of deviation valor between the three samples analyzed at each point
Section 3.1.3
Figure 4 should indicate the range of deviation valor between the three samples analyzed at each point
Section 3.1.4
Figure 5 should indicate the range of deviation valor between the three samples analyzed at each point
Section 3.1.5
The levels indicated in line 308 “0.8%, 1.6%, 2.4%, 3.2%, 4.0%, 5.6%, and 6.4%. “ do not coincide with the levels indicated in line 140 “(0.8%, 1.6%, 2.4%, 3.2%, 4.0%, 4.8%, 5.6%, 6.4%)”
Figure 6 should indicate the range of deviation value between the three samples analyzed at each point
Table 4, include for each source include the % contribution and the value of p
Section 3.2.2
To complete the BB design, you will need to include the regression polynomial, canonical analysis to determine the stationary point, and the predicted response.
The discussion of the results obtained against other research should be carried out in order to assess the contribution achieved

Section 3.3
In the description of the results in this section, numerical values should be included and compared against other research in order to assess the contribution achieved
Section 3.4
The discussion of the results obtained against other research should be carried out in order to assess the contribution achieved
Section .35
The discussion of the results obtained against other research should be carried out in order to assess the contribution achieved

Additional comments

Authors are requested to address ALL the observations made, so that the article is structured in an appropriate manner.

---

## Round 0.3 · Minor Revisions

After reviewing the manuscript some issues arise. Please provide the retention index values ​​(theoretical and practical) for the volatile compounds. Please note that caffeine is not a volatile compound and therefore should not be detected by the methodology used.

·

Basic reporting

No comment

Experimental design

No comment

Validity of the findings

No comment

·

Basic reporting

The authors have addressed the remarks made. It meets the requirements to be considered for publication.

Experimental design

The experimental design is appropriate, and the observations made have been addressed.

Validity of the findings

The validations carried out that support the developed research are considered adequate.

Additional comments

none

---

## Round 0.4 · accepted · Accept

The reviewers recommended publishing the articles after the corrections were made.